# Learning from Few Samples with Language-Model Guidance

## Abstract

We consider the problem of learning a classifier from a small set of high-dimensional datapoints, with access to domain knowledge from a language model or human expert. How should such domain knowledge be elicited and leveraged together with the data? We investigate four strategies and their combinations, for linear models, the de facto choice in the small-sample regime: (1) feature selection, (2) coefficient sign constraints, (3) inequality constraints on coefficient magnitudes, and (4) equality constraints across related features. Each restricts the hypothesis class, improving generalization in small sample settings, and corresponds to a natural query for an expert or Large Language Model (LLM) that draws on prior knowledge. We evaluate these approaches on four clinical prediction tasks involving high-dimensional molecular measurements, where labeled datasets are typically limited to small patient cohorts. We find models trained with as few as five patients outperform baselines requiring ten times more patients, with improvements upwards of 20% AUC, exceeding human expert–designed models. This highlights new strategies for LLM-guided model design in data-limited settings.

## 1 Introduction

Suppose that you are a clinician studying a small cohort of patients with a rare disease. For each patient, you have access to thousands of molecular measurements, such as gene expression or protein abundance levels, and you seek to develop a predictive model for an important clinical outcome or diagnosis. One natural approach is to train a simple class of models, relying on high regularization to mitigate the severe data limitations (Santos & Papa, 2022). Alternatively, you might depend entirely on clinical expertise or domain knowledge to manually construct a model, sidestepping the data altogether.

Figure 1 provides a concrete illustration of such a setting. Here, the task is predicting postoperative surgical site infections (SSI) using a 721-protein omic dataset collected from preoperative blood samples (Hédou et al., 2024). The right endpoint of each plot represents the performance of a purely knowledge-driven linear model, whose weights are provided by a large language model (LLM) without access to any patient data. To obtain the weights, the LLM was given a single prompt with the list of 721 genes, and asked to assign a real-valued regression weight reflecting each gene's expected association with SSI risk. The left endpoint shows the performance of a purely data-driven linear model, trained with standard $\ell_2$ regularization but no domain knowledge. Between these extremes, we also show a naive interpolation: training a linear model with an objective that balances two $\ell_2$ regularization terms: one encouraging weights close to zero (purely data-driven) and another pulling weights towards the LLM-provided model (purely knowledge-driven).

We observe two key phenomena. First, at least for the small datasets of $n = 10$ patients, the purely knowledge-driven (LLM) model consistently outperforms the purely data-driven ($\ell_2$ regularized) model *for all regularization strengths* (Hoerl & Kennard, 1970b). Second, in the low-data setting, there exists an intermediate regime where combining both knowledge and data is strictly better than either endpoint.

These observations prompt two fundamental questions: *What are the most effective strategies for combining domain knowledge with data? And under which conditions or parameter regimes should we expect to surpass the performance achievable by either knowledge or data alone?*

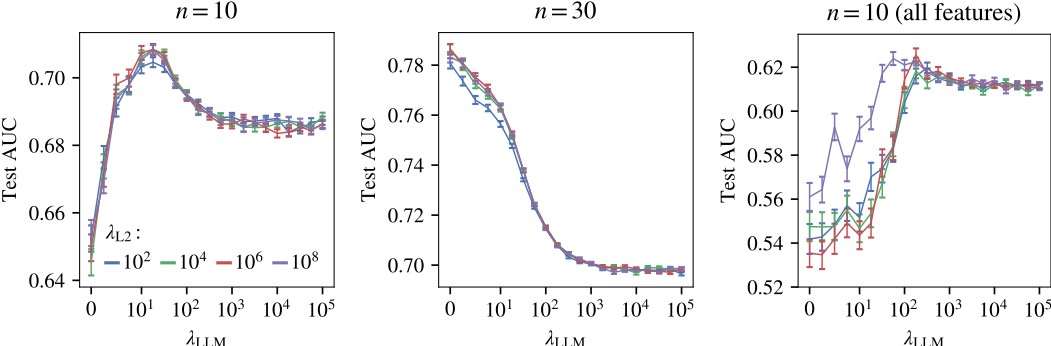

Figure 1: **Interpolation between data-derived model and LLM-specified model.** We prompted an LLM for coefficient values $w_{\mathrm{LLM}}$ on the 721-protein SSI dataset and fit a logistic model $\min_w \mathcal{L}_{\log}(w) + \lambda_{\mathrm{LLM}}\|w - w_{\mathrm{LLM}}\|_2^2 + \lambda_{\mathrm{L2}}\|w\|_2^2$ that interpolates between the purely data-derived model (left endpoint) and the LLM-specified model (right endpoint). Error bars denote $\pm$ standard error of the mean over 2000 splits. **Left:** $n = 10$ training examples, 20 features whose LLM coefficients have the largest magnitudes. **Center:** $n = 30$, the same 20 features. **Right:** $n = 10$, all 721 features. In the small data regime ($n = 10$), the LLM-specified model outperforms the data-derived models across $\ell_2$ regularization strengths, whereas the opposite is the case for $n = 30$. For $n = 10$, for both $d = 10$ and $d = 721$, there is a choice of regularization parameter for which leveraging both data and the LLM outperforms either alone.

These classic questions have gained renewed interest with the rise of large language models (LLMs) (Zhang et al., 2024; Vaswani et al., 2017; Minaee et al., 2024). LLMs promise to assimilate vast amounts of domain knowledge and deliver it in customized forms based on user queries (Lee et al., 2024). While the reliability of responses may be varied, their scale is unprecedented (Hendrycks et al., 2020; Naveed et al., 2023). LLMs provide an unparalleled resource for articulating domain knowledge in as many nuanced forms as users can suggest. Thus, it becomes essential to revisit the fundamental problem: how best can we train predictive models in this new paradigm, where rich domain expertise is abundant and can be elicited and represented in diverse ways.

This question is especially pressing for small-sample, high-dimensional datasets, ubiquitous across clinical and scientific domains. In medicine, while many measurements can be made for a single patient, assembling large patient cohorts is often constrained by cost and logistics, and may be infeasible for rare diseases and phenotypes (Mitani & Haneuse, 2020; Banerjee et al., 2023a). In molecular biology, assays can measure thousands of features per biological sample but the number of samples may be limited to dozens or fewer (Krawczuk & Łukaszuk, 2016). In these settings even the simplest linear models overfit and fail to generalize, and higher-capacity approaches including deep learning exacerbate these issues (Berisha et al., 2021; Alsentzer et al., 2025). Developing methods that learn reliably from a small number of high-dimensional datapoints remains a fundamental challenge.

## 2 METHODS

In this work, we focus on the severely data-limited setting, with train set sizes of no more than $n = 50$, and as small as $n = 5$. This choice has two motivations. First, this is a clinically and scientifically relevant and underexplored regime, that likely has the most to gain from leveraging domain knowledge (Banerjee et al., 2023b). Second, this regime allows us to evaluate our techniques with statistical validity given even modest sized datasets, by evaluating performance across independently drawn subsets of $n$ datapoints. We study linear models since their sample efficiency makes them appropriate for the setting we consider of extremely small datasets, and because, as we show, they allow us to transparently incorporate domain knowledge into the model.

The key idea of our approach is to encode elicited domain knowledge as constraints on the hypothesis class. These constraints define the feasible region for the model parameters, ensuring that the learned model reflects both data and prior knowledge. By reducing the size of the hypothesis class, constraints improve sample complexity and generalization in small-sample settings.

### 2.1 ELICITING USEFUL INFORMATION

We explore four types of domain knowledge, each designed to (1) be easily elicited from either an LLM or domain expert, (2) admit a natural interpretation as a constraint on the parameters of a

linear model, (3) yield constraint formulations that permit efficient optimization, and (4) restrict the hypothesis class to improve generalization. We elicit and evaluate the following constraints.

**Zero constraints.** We ask the expert or LLM to nominates a shortlist of features deemed most relevant to the task. We restrict the model to these features, fixing all other coefficients to zero. This directly addresses the challenges of high-dimensional, low-sample settings by focusing the model on a small, curated set of predictors.

**Group constraints.** High-dimensional datasets often contain redundant features that reflect the same latent factor. We ask the expert or LLM to identify such groups and constrain features within each group to share a common weight (after standardization). Grouping reduces dimensionality and mitigates measurement noise by averaging across features.

**Sign constraints.** Given a set of features, we elicit the expected sign of the coefficient for each one, based on knowledge of the direction of effect. This restricts the model's weight vector to an orthant, reducing the hypothesis space.

**Inequality Constraints.** Given a set of features, we ask the expert or LLM to identify the most important ones. We then require that the smallest coefficient magnitude among these features is at least as large as the largest magnitude among all others. This encodes relative importance and restricts the hypothesis class to an intersection of half-spaces.

All constraint types, and any subset, can be combined within a single model. This composability enables multiple forms of domain knowledge to be jointly leveraged. We elicit this information from either an LLM or a human domain expert using a prompt that includes a description of the constraint type, along with information about the dataset, classification task, and modeling approach, which we describe in the sections that follow. For each dataset, we prompt separately for each constraint type, requesting both an answer and a brief explanation. All results in the main text use GPT-4o, because it does not perform internet search and therefore offers a clearer characterization of intrinsic LLM domain knowledge. Appendix A reports results with other LLMs. Appendix D contains example prompts used for the LLM and domain expert.

## 2.2 MODEL FORMULATION

Given the elicited information, we train a binary classifier $f(x) = w^\top x + b$, where $x \in \mathbb{R}^d$ is a standardized feature vector, $w \in \mathbb{R}^d$ is the learned weight vector, and $b \in \mathbb{R}$ is a bias term. Given training data $\{(x_i, y_i)\}_{i=1}^n$ with labels $y_i \in \{0, 1\}$, we minimize a class-balanced, regularized logistic loss subject to a constraint set $\mathcal{C} \subseteq \mathbb{R}^d$:

$$\min_{w \in \mathcal{C}, b \in \mathbb{R}} \frac{1}{n} \sum_{i=1}^n \alpha_{y_i} \left[ \log(1 + e^{w^\top x_i + b}) - y_i(w^\top x_i + b) \right] + \lambda \|w\|_2^2,$$

where $\lambda \geq 0$ the strength of L2 regularization and $\alpha_{y_i} = (\sum_{j=1}^n \mathbf{1}\{y_j = y_i\})^{-1}$ balances the contribution of each class to the loss. The feasible region is specified by $\mathcal{C} \subseteq \mathbb{R}^d$ encoding domain knowledge through four classes of constraints. Feature selection is encoded by a zeroing set $Z \subseteq [d]$, requiring $w_j = 0$ for all $j \in Z$. Group constraints are expressed by a collection of disjoint index sets $G = \{G_1, G_2, \ldots\}$, with all features in each group constrained to share a common weight: $w_j = w_k$ for all $j, k \in G_\ell$. Sign constraints are specified by a set $S = \{(j, s_j)\}$ with $s_j \in \{-1, 1\}$, imposing $s_j w_j \geq 0$ for each indexed feature. Feature importance constraints are defined by a set $I \subseteq [d]$ of features designated as more important than the rest. The model is required to assign at least as much magnitude to each feature in $I$ as to any feature not in $I$, which imposes the constraint $|w_j| \geq |w_k|$ for all $j \in I$, $k \notin I$. When sign constraints are also specified for both $j$ and $k$, the constraint becomes linear: $s_j w_j \geq s_k w_k$. The constraint set $\mathcal{C}$ restricts the model to a limited subset of features and directly controls sparsity, so we employ an $\ell_2$ regularizer rather than a sparsity-promoting penalty such as $\ell_1$.

The full constraint set is denoted $\mathcal{C} = Z \cup G \cup S \cup I$, though any subset may be used in practice. All combinations of these constraints yield a convex feasible region, except when inequality constraints $I$ are imposed without corresponding sign constraints $S$. In the convex case, the objective is efficiently solvable using standard convex optimization methods. All constrained models are specified and solved using CVXPY (Diamond & Boyd, 2016). For simplicity, we interchangeably use non-italicized constrain set variables and "+" in place of "$\cup$" throughout the paper.

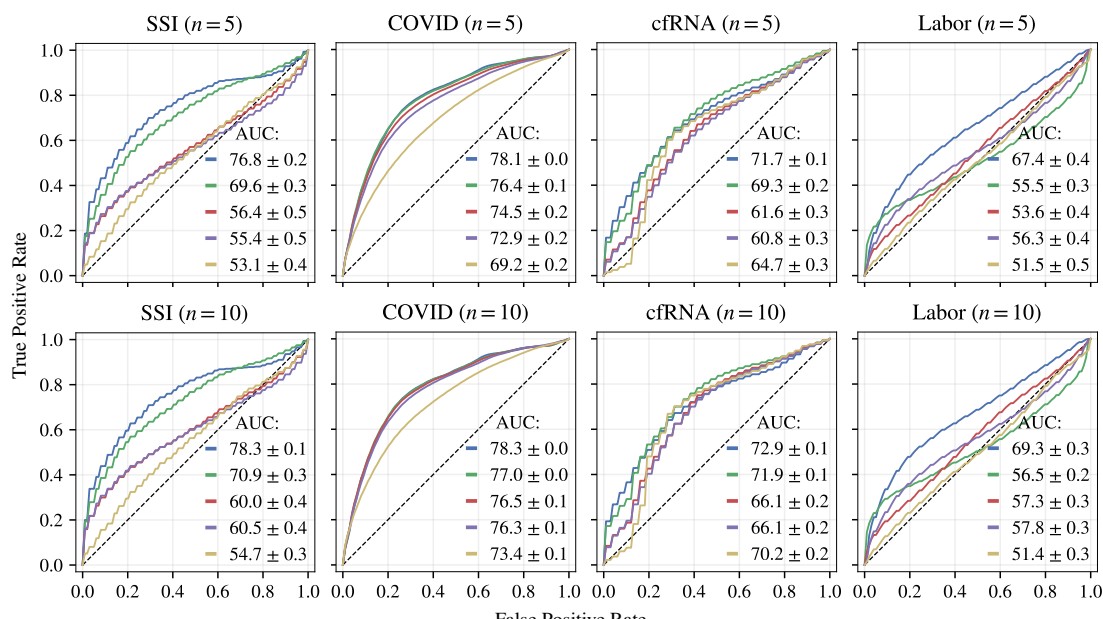

Figure 2: **LLM constraints substantially improve generalization in small-sample settings.** Each panel shows the mean area under the receiver-operator curve (AUC) over 2000 random splits for logistic models trained under five constraint regimes. Shaded areas, narrower than the line thickness, indicate $\pm$ standard error of the mean. **Columns:** Four clinical omics datasets (SSI, COVID, cfRNA, Labor). **Top row:** $n = 5$ training examples. **Bottom row:** $n = 10$. AUC curves for $n = 20$ and $n = 50$ are in Appendix Figure 1. Models include: Ridge (no constraints), Z only, Z+G, Z+G+S, Z+G+S+I. The full set (Z+G+S+I) achieves highest AUC on every dataset when $n = 5$ and $n = 10$, indicating that constraints are most valuable in the most sample-limited regimes. In some cases, constrained models trained on just 5 samples approach or exceed the performance of unconstrained models trained on the full dataset (Table 1). Combining constraints improves performance, suggesting that constraint types add complementary inductive bias. This highlights the promise of LLM-elicited constraints in low-$n$, high-$d$ settings.

## 2.3 DATASETS AND SAMPLING

We selected four clinically relevant omic datasets that exemplify the high-dimensional, small-sample regime typical in precision medicine. In each dataset, samples represent patient-derived biological measurements, with features corresponding to the abundance of hundreds to thousands of molecular markers. These datasets span molecular modalities, including proteomics and cell-free RNA, and target prediction tasks with clinical relevance.

Collecting data on large patient cohorts remains expensive and logistically challenging, particularly for omic assays that require specialized protocols and longitudinal sampling. In many clinical contexts, especially those involving rare diseases and phenotypes, assembling large labeled datasets is infeasible. The datasets we study are notable because they represent rare instances where such high-dimensional measurements have been obtained at cohort scale. Yet even so, the number of labeled examples remains small by modern machine learning standards.

**SSI.** Consists of preoperative blood samples from 91 adults undergoing elective colorectal surgery, each measuring the abundance of 712 proteins. The task is to predict the occurrence of surgical site infection (SSI), which could inform targeted antibiotic prophylaxis and reduce postoperative complications (Hédou et al., 2024).

**COVID.** Comprises blood samples from 68 patients with confirmed COVID-19, each measuring the abundance of 1,472 proteins. The task is to predict disease severity (mild/moderate vs. severe). An independent patient cohort includes 784 samples from 306 additional patients. Early identification of high-risk patients could inform appropriate triage and therapeutic intervention (Filbin et al., 2021).

**cfRNA.** Contains longitudinal blood samples from 49 pregnant women (29 who developed preeclampsia and 20 controls), collected at multiple time points prior to clinical diagnosis. Each sample contains

abundance measurements for 37,184 cell-free RNA transcripts. Preeclampsia is a leading cause of maternal and fetal morbidity, and early prediction could guide preemptive management (Moufarrej et al., 2022; Marić et al., 2022).

**Labor.** Includes plasma proteomic profiles from 63 pregnant women, each measuring the abundance of 1,317 proteins. The prediction task is labor onset, framed as binary classification of preterm birth by thresholding gestational age at 37 weeks. Anticipating preterm birth is critical for managing delivery and preventing neonatal complications (Stelzer et al., 2021).

The COVID dataset includes a designated held-out cohort measured in an independent study. For the SSI, cfRNA, and Labor datasets, patients are randomly split 50/50 into a held-out evaluation set (for Test AUC) and a training pool. For each target sample size $n$, we draw uniform random subsets of size $n$ from the pool, train models on these subsets, and report performance on the evaluation set averaged over subsamples. To ensure class coverage, we evaluate models only on subsamples that contain at least one datapoint from each class. The datasets exhibit class imbalance; notably, in the Labor dataset, only two preterm birth cases may appear among the training examples. Full details on the datasets, preprocessing, and model training are given in Appendix B and Appendix C.

## 3 RESULTS

LLM-derived constraints substantially improve performance in small-sample settings, with gains exceeding 20% AUC, and models trained on $n = 5$ outperforming baselines trained on ten times more data. Comparison of different combinations of constraint types shows that each contributes complementary value, with the full set (Z+G+S+I) achieving the best results at low $n$ across all but one dataset. Constraints also reduce performance variability across train–test splits, illustrating enhanced reliability when training on a single small training set. LLM-derived constraints significantly

Table 1: **Performance across methods, datasets, and number of samples.** Test AUC (%) reported as mean ± standard error with 90% interpercentile range [5th–95th percentile] over 500 splits for LLM methods and 100 splits for the remaining methods. Methods include all convex combinations of LLM-provided constraints (Z, G, S, I), linear baselines (Lasso, Ridge, RFE, MRMR, MI), and two more recent LLM-based methods (Jeong, Zhang). The Labor dataset contains only two positive-class examples in the pool from which training sets are drawn, limiting the statistical validity of conclusions drawn from this dataset. LLM-constrained models outperform baselines across datasets and sample sizes, with gains in AUC of $> 20\%$ in some instances. Performance improves as more constraints are added, and the full set performs best in many cases. Further, the interpercentile range decreases, highlighting improved reliability when fitting models on a single training set.

| $n$ | Z+G+S+I | Z+G+S | Z+S+I | Z+G | Z+S | Z | Lasso | Ridge | RFE | MRMR | MI | Jeong | Zhang |
|---|---|---|---|---|---|---|---|---|---|---|---|---|---|
| **SSI** | | | | | | | | | | | | | |
| 5 | 76.8±0.3 [62.9–84.9] | 69.6±0.3 [47.4–82.3] | 73.0±0.5 [57.3–84.7] | 56.4±0.5 [24.3–79.4] | 68.5±0.7 [45.4–85.5] | 55.4±0.5 [27.0–81.6] | 57.4±5.5 [41.4–74.3] | 53.1±0.4 [32.9–74.3] | 48.8±1.8 [29.9–69.3] | 47.4±1.8 [26.4–65.6] | 52.8±2.1 [32.3–76.4] | 56.5±2.0 [35.9–80.0] | 54.8±1.9 [34.3–75.5] |
| 10 | 78.3±0.1 [65.9–83.2] | 70.9±0.3 [53.0–81.2] | 74.0±0.4 [58.6–83.9] | 60.0±0.4 [29.4–81.9] | 71.5±0.5 [49.0–86.8] | 60.5±0.4 [30.0–85.9] | 51.9±1.9 [30.9–66.2] | 54.7±.3 [27.6–76.7] | 52.7±1.6 [36.1–70.9] | 55.0±0.9 [33.9–70.7] | 53.0±2.1 [28.0–72.9] | 62.3±2.0 [41.9–83.3] | 52.3±2.5 [31.0–72.4] |
| 20 | 78.9±0.2 [73.4–83.9] | 74.6±0.3 [60.9–83.9] | 77.4±0.3 [65.7–85.9] | 68.5±0.5 [44.4–84.5] | 75.9±0.4 [56.8–87.2] | 70.1±0.7 [39.5–87.2] | 58.6±2.6 [46.3–72.9] | 59.3±1.4 [34.4–75.1] | 52.6±1.2 [34.0–75.6] | 55.5±1.8 [38.9–75.7] | 55.5±1.7 [38.9–75.7] | 65.9±1.9 [42.8–85.1] | 58.8±2.5 [46.3–75.5] |
| 50 | 79.6±0.0 [79.6–79.6] | 76.3±0.0 [76.3–76.3] | 81.6±0.0 [81.6–81.6] | 76.0±0.0 [76.0–76.0] | 81.9±0.0 [81.9–81.9] | 85.5±0.0 [85.5–85.5] | 66.6±3.1 [55.9–74.9] | 73.0±0.0 [73.0–73.0] | 56.0±0.8 [45.7–65.1] | 52.7±0.8 [43.0–59.5] | 52.7±0.8 [42.7–60.3] | 74.8±0.7 [66.4–80.1] | 66.1±2.7 [56.3–74.6] |
| **COVID** | | | | | | | | | | | | | |
| 5 | 78.1±0.0 [75.3–79.0] | 76.4±0.1 [72.7–79.0] | 77.1±0.1 [74.8–79.0] | 74.5±0.2 [63.0–79.0] | 76.1±0.1 [72.6–79.1] | 72.9±0.2 [44.3–78.8] | 60.8±2.1 [43.1–72.8] | 69.2±0.2 [50.0–79.8] | 65.5±1.4 [45.6–76.3] | 64.3±1.3 [49.5–73.5] | 62.9±1.6 [37.3–73.7] | 61.1±1.4 [42.7–73.4] | 61.7±2.1 [49.7–73.6] |
| 10 | 78.3±0.0 [76.1–78.7] | 77.0±0.0 [74.6–79.1] | 77.5±0.0 [76.1–78.8] | 77.2±0.0 [74.2–79.0] | 77.1±0.1 [74.7–79.0] | 76.3±0.1 [73.5–79.2] | 72.5±2.4 [63.9–77.9] | 73.4±0.1 [64.9–79.2] | 73.2±0.6 [65.7–78.7] | 73.0±0.3 [68.7–76.4] | 73.0±0.4 [67.9–77.0] | 69.2±0.9 [57.9–77.2] | 68.8±1.5 [63.5–73.1] |
| 20 | 77.4±0.0 [76.4–78.5] | 77.4±0.0 [76.0–78.8] | 77.8±0.0 [76.9–78.6] | 77.2±0.0 [75.7–78.8] | 77.6±0.0 [76.2–78.8] | 77.8±0.0 [76.5–79.2] | 72.5±2.4 [63.9–77.9] | 75.7±0.4 [66.9–79.3] | 74.6±0.6 [68.4–78.9] | 73.3±0.3 [69.7–76.4] | 73.9±0.3 [70.3–76.7] | 69.2±0.9 [57.9–77.2] | 68.8±1.5 [63.5–73.1] |
| 50 | 77.5±0.0 [77.1–77.9] | 77.6±0.0 [77.0–78.1] | 78.0±0.0 [77.6–78.4] | 77.6±0.0 [77.0–78.1] | 78.0±0.0 [77.4–78.5] | 78.2±0.0 [77.6–78.8] | 75.2±0.3 [73.2–77.5] | 78.3±0.1 [77.4–79.0] | 76.7±0.3 [74.3–80.2] | 74.1±0.2 [72.4–75.5] | 74.0±0.2 [71.8–76.3] | 76.0±0.4 [70.5–78.6] | 74.4±0.4 [69.7–77.9] |
| **cfRNA** | | | | | | | | | | | | | |
| 5 | 71.7±0.1 [59.8–77.1] | 69.3±0.2 [49.8–76.9] | 66.0±0.3 [35.3–73.6] | 61.6±0.3 [35.3–73.6] | 64.1±0.5 [41.0–77.0] | 60.8±0.3 [36.1–74.4] | 58.9±2.7 [46.3–68.5] | 64.7±0.3 [36.7–74.6] | 65.1±1.5 [48.0–76.3] | 60.4±4.1 [31.9–74.0] | 66.7±1.9 [61.4–72.8] | 65.4±3.0 [56.5–73.7] | 61.7±1.2 [58.0–65.9] |
| 10 | 72.9±0.1 [66.5–77.2] | 71.9±0.1 [61.5–78.3] | 68.3±0.3 [56.3–76.4] | 66.1±0.2 [44.8–75.7] | 69.0±0.4 [52.1–78.1] | 66.1±0.2 [45.4–76.8] | 63.4±1.8 [47.5–78.9] | 70.2±0.2 [39.4–76.3] | 67.8±1.3 [48.5–77.4] | 66.1±1.3 [47.1–76.3] | 66.0±1.2 [51.5–76.5] | 63.6±1.2 [47.6–71.1] | 63.1±4.7 [44.3–78.5] |
| 20 | 72.8±0.1 [68.9–77.3] | 73.7±0.1 [68.9–78.8] | 72.1±0.2 [64.9–77.8] | 69.5±0.2 [60.1–76.8] | 73.4±0.2 [62.7–80.0] | 70.6±0.2 [62.3–78.1] | 71.8±2.2 [55.7–82.3] | 70.4±0.5 [60.2–77.8] | 75.0±1.1 [63.1–87.4] | 66.1±0.8 [60.1–78.9] | 71.2±0.7 [60.1–78.7] | 66.2±0.8 [55.4–74.3] | 71.2±3.6 [59.1–86.5] |
| 50 | 73.0±0.1 [69.5–76.6] | 74.5±0.1 [70.8–77.5] | 75.8±0.1 [72.2–78.6] | 70.7±0.1 [67.8–75.4] | 77.2±0.1 [73.4–80.4] | 73.7±0.1 [70.8–77.8] | 80.0±1.1 [72.0–88.8] | 74.4±0.2 [71.4–78.6] | 82.1±0.6 [75.9–87.3] | 76.5±0.6 [70.5–81.7] | 76.0±0.7 [68.9–84.4] | 69.8±0.5 [64.9–75.8] | 78.3±1.7 [71.7–85.9] |
| **Labor** | | | | | | | | | | | | | |
| 5 | 67.4±0.4 [61.1–72.2] | 55.5±0.3 [51.4–58.5] | 54.1±0.5 [50.2–61.3] | 53.6±0.4 [51.1–61.2] | 52.1±0.4 [50.2–65.4] | 51.5±0.5 [50.1–66.5] | 53.9±2.5 [26.2–80.3] | 60.3±0.4 [50.0–63.4] | 50.2±1.7 [33.8–70.5] | 52.1±2.4 [24.1–77.4] | 49.9±2.0 [24.4–71.1] | 53.6±0.5 [45.1–54.9] | 53.9±2.5 [26.2–80.3] |
| 10 | 69.3±0.3 [22.1–70.3] | 56.5±0.2 [50.0–63.4] | 57.3±0.3 [33.8–70.5] | 57.3±0.3 [33.8–70.5] | 52.1±2.4 [24.1–77.4] | 57.8±0.3 [69.6–85.0] | 55.1±4.9 [31.9–74.0] | 62.1±0.2 [50.0–68.8] | 50.5±1.6 [34.1–70.5] | 49.6±2.0 [26.7–66.3] | 50.9±2.2 [26.7–66.3] | 54.7±1.7 [36.7–76.9] | 56.3±4.1 [37.2–72.4] |
| 20 | 79.5±0.1 [75.3–82.8] | 77.9±0.1 [74.0–81.5] | 78.4±0.1 [73.5–84.2] | 77.8±0.3 [69.8–85.5] | 78.9±0.2 [74.4–83.8] | 77.1±0.3 [70.3–84.7] | 62.2±3.1 [49.4–69.1] | 57.7±0.8 [50.0–68.6] | 52.5±1.4 [34.6–66.5] | 54.6±2.1 [33.3–78.7] | 54.6±2.1 [29.7–81.5] | 54.7±0.2 [54.9–54.9] | 55.4±2.2 [43.7–70.7] |
| 50 | 81.3±0.1 [78.2–82.9] | 80.5±0.1 [77.6–82.5] | 80.1±0.1 [75.1–84.2] | 79.1±0.1 [75.8–82.8] | 81.3±0.1 [77.9–84.4] | 77.0±0.1 [71.3–81.4] | 69.7±2.8 [58.5–85.9] | 65.8±0.6 [50.0–71.7] | 57.6±2.0 [33.2–77.2] | 59.6±2.1 [31.3–79.1] | 59.7±2.0 [33.0–78.9] | 52.0±1.2 [41.1–70.0] | 67.2±1.9 [52.0–78.3] |

outperform constraints elicited from a human expert and from constraints derived from correlation information between features and labels in the training set. Performance remains robust to changes in LLM version, prompt phrasing, and regularization strength. Together, these results highlight the utility of LLM-elicited structure in high-$d$, low-$n$ classification settings.

**Constrained models generalize with as few as five patients.** Table 1 and Figures 2 and 4 present the performance of our constrained models alongside a range of baseline methods: LASSO (Tibshirani, 1996b), Ridge (Hoerl & Kennard, 1970b), Recursive Feature Elimination (RFE) (Guyon et al., 2002b), Maximum Relevance Minimum Redundancy (MRMR) (Ding & Peng, 2005b), Mutual Information (MI) (Verleysen et al., 2009), LLM-Select (Jeong et al., 2024), and LLM-Lasso (Zhang et al., 2025). We additionally compare to nonparametric models ($k$-nearest neighbors, naive Bayes, decision trees, random forests) and neural networks (multi-layer perceptrons with one or two hidden layers of 10, 20, or 50 ReLU units), in Appendix Figures 5 and 6.

Our constrained models achieve performance surpassing existing methods, even when trained with ten times less data. On the SSI dataset, the fully constrained model (Z+G+S+I) achieves a test AUC of 76.8% with 5 training samples, outperforming all baselines trained on up to 50 samples. Our performance increases with more data, up to 79.6% with 50 training samples. On COVID, our fully constrained approach reaches 78.1% with $n = 5$, surpassing the best baseline with the same training size, and closely matching the best method trained on 50 samples, Ridge (78.3%). On cfRNA, the fully constrained model attains 71.7% with $n = 5$, outperforming all baselines trained up to $n = 10$. At larger training sizes $n = 20$ and $n = 50$, RFE and Lasso outperform the constrained models. All other baselines perform worse than our approach up to $n = 50$. On Labor, the fully constrained model attains 76.4% AUC with $n = 5$, over a 22% gain in AUC over the best baseline with $n = 5$. The constrained model trained with $n = 5$ samples exceeds all other baselines, even when the baselines have access to $n = 50$ samples. Across datasets, at the most extreme data-limited setting, the fully contrained model attains significant performance gains. For most datasets, as the training set increases in size, the performance of the constrained models increases moderately, still maintaining a lead over baseline methods, but the performance gains shrink.

All baselines perform poorly in the extreme data-limited regime, with AUCs typically below 60% at $n = 5$, and gaps of up to 20% AUC between our method and the best baseline. This underscores the absence of any effective approaches in this regime. Across datasets, no model class reliably outperforms linear models in the most data-limited settings of $n = 5$ and $n = 10$. More complex models, including the MLP methods, show improvement at larger $n$, but underperform in low-$n$ regimes, supporting our focus on constrained linear models in this sample-limited setting.

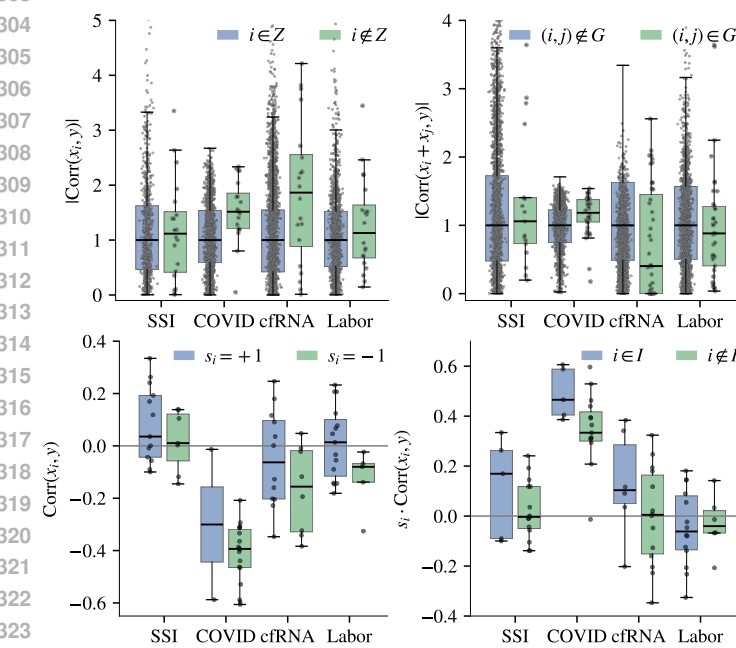

Figure 3: **LLM constraints compared to feature-label and feature-feature correlation.** Each panel shows correlation between features and labels stratified by LLM-proposed constraint sets. **Top left:** Features in the zeroing set $Z$ tend to show higher absolute correlation with the label. **Top right:** Grouped feature pairs $G$ show stronger mutual correlation, consistent with redundancy. **Bottom left:** Features with sign constraints match the direction of correlation. **Bottom right:** Features marked important ($I$) exhibit larger signed correlations. Proposed constraints sometimes deviate from empirical correlations, potentially reflecting corpus gaps, noise in correlation estimates, or an LLM strategy that evaluates constraints together rather than individually. This suggests that constrained models can perform well even when some LLM-elicited domain knowledge is incorrect or inconsistent with trends in the data.

**Constraint types contribute complementary benefit.** To assess the contributions of each constraint type, individually and in combination, we evaluate all convex combinations of the four constraint types: Z+G+S+I, Z+G+S, Z+S+I, Z+G, Z+S, Z (Table 1 and Figures 2 and 4). This allows us to study performance as constraints are composed together, and further, constitutes an ablation study. Feature selection alone (Z) improves test AUC relative to the unconstrained model (Ridge) across almost all datasets and training set sizes, with the exception of the cfRNA dataset. This dataset has far more features than the others (approx. 30k vs. 1k), so this may reflect difficulty encountered by the LLM in reasoning over a large feature set.

Adding sign constraints (Z+S) improves or maintains performance across all datasets and is similar to the effect of adding group constraints (Z+G), as well as their combination (Z+G+S). The consistency of these improvements may arise from the LLM assigning directional constraints only to features it has already selected, for which it likely possesses higher-confidence prior knowledge, and from the fact that grouped genes and proteins may be tightly co-regulated across biological conditions throughout the scien-

Table 2: **Comparison to correlation-based constraints.** Test AUC (%) reported as mean ± standard error, [5th–95th percentile] over 2000 random splits with $n = 5$, for all convex subsets of constraints (columns). Constraints proposed by the LLM are compared to those built using empirical feature–label and feature-feature correlations from either the same training data (CORR. ($n = 5$)), all available data (CORR. ($n =$ all)), or random selection. Across datasets, LLM constraints outperform those based on the training set alone, and approach the performance of constraints built using the full dataset.

| | Z+G+I+S | Z+G+S | Z+S+I | Z+G | Z+S | Z |
|---|---|---|---|---|---|---|
| **SSI** | | | | | | |
| LLM | 76.8±0.3 [62.9–84.9] | 69.6±0.3 [47.4–82.3] | 73.0±0.5 [57.3–84.7] | 56.4±0.5 [24.3–79.4] | 68.5±0.7 [45.4–85.5] | 55.4±0.5 [27.0–81.6] |
| CORR. ($n = 5$) | 65.1±2.9 [36.2–83.6] | 59.5±3.0 [30.4–82.7] | 65.1±2.9 [36.2–83.6] | 55.3±2.4 [32.2–78.5] | 57.7±2.8 [28.6–84.2] | 52.7±2.6 [27.4–77.9] |
| CORR. ($n =$ all) | 79.3±2.1 [61.2–85.8] | 77.4±2.1 [62.3–85.9] | 79.3±2.1 [61.2–85.8] | 76.5±2.0 [58.6–82.1] | 76.6±2.0 [61.9–85.7] | 54.6±2.3 [41.8–67.2] |
| Random Selection | 67.3±3.1 [46.7–83.9] | 62.4±2.7 [43.5–79.0] | 67.3±3.1 [46.7–83.9] | 60.8±2.8 [39.1–78.1] | 74.3±2.0 [60.0–84.4] | 51.2±2.5 [29.7–72.1] |
| **COVID** | | | | | | |
| LLM | 78.1±0.0 [75.3–79.0] | 76.4±0.1 [72.7–79.0] | 77.1±0.1 [74.8–79.0] | 74.5±0.2 [63.0–79.0] | 76.1±0.1 [72.6–79.1] | 72.9±0.2 [44.3–78.8] |
| CORR. ($n = 5$) | 75.4±2.2 [63.2–78.4] | 71.1±2.4 [59.7–76.9] | 75.4±2.2 [63.2–78.4] | 67.4±2.5 [51.9–75.0] | 72.4±2.0 [60.1–76.8] | 63.1±2.6 [45.3–74.7] |
| CORR. ($n =$ all) | 76.2±1.5 [69.4–79.0] | 74.1±1.6 [67.3–78.6] | 76.2±1.5 [69.4–79.0] | 70.4±1.8 [62.3–76.6] | 76.4±1.2 [71.5–78.7] | 73.1±1.1 [67.4–78.2] |
| Random Selection | 76.1±2.0 [60.5–78.4] | 70.5±2.3 [59.3–77.0] | 76.1±2.0 [60.5–78.4] | 66.3±2.8 [52.0–74.6] | 71.1±1.9 [60.8–76.3] | 62.1±2.4 [47.5–72.8] |
| **cfRNA** | | | | | | |
| LLM | 71.7±0.1 [59.8–77.1] | 69.3±0.2 [49.8–76.9] | 66.0±0.3 [51.1–75.5] | 61.6±0.3 [35.3–73.6] | 64.1±0.5 [41.0–77.0] | 60.8±0.3 [36.1–74.4] |
| CORR. ($n = 5$) | 64.8±2.3 [45.4–77.1] | 59.0±2.4 [43.1–74.6] | 64.8±2.3 [45.4–77.1] | 55.6±2.8 [33.9–71.2] | 61.3±2.2 [36.1–74.3] | 58.5±2.5 [35.5–71.0] |
| CORR. ($n =$ all) | 68.6±2.0 [43.8–78.5] | 69.5±2.1 [42.4–77.2] | 68.6±2.0 [43.8–78.5] | 62.2±2.3 [39.8–74.9] | 68.4±1.8 [47.1–77.6] | 67.6±1.7 [44.4–75.9] |
| Random Selection | 62.9±2.7 [43.7–75.7] | 57.8±2.4 [40.9–72.0] | 62.9±2.7 [43.7–75.7] | 54.2±2.6 [35.3–69.8] | 54.4±2.3 [36.7–70.1] | 54.2±2.2 [33.0–68.2] |
| **Labor** | | | | | | |
| LLM | 67.4±0.4 [61.1–72.2] | 55.5±0.3 [51.4–58.5] | 54.1±0.5 [50.2–61.3] | 53.6±0.4 [51.1–61.2] | 52.1±0.4 [50.2–65.4] | 51.5±0.5 [50.1–66.5] |
| CORR. ($n = 5$) | 64.4±2.1 [50.1–71.9] | 60.3±2.5 [46.7–70.4] | 64.4±2.1 [50.1–71.9] | 57.9±2.4 [44.4–71.2] | 63.8±2.3 [52.8–76.2] | 42.1±2.9 [23.5–60.9] |
| CORR. ($n =$ all) | 66.2±1.2 [58.4–72.4] | 64.5±1.3 [56.9–71.7] | 66.2±1.2 [58.4–72.4] | 62.1±1.5 [54.2–70.2] | 65.5±1.1 [58.1–72.3] | 68.9±1.0 [62.2–75.1] |
| Random Selection | 54.5±2.6 [34.3–68.2] | 52.1±2.5 [33.8–66.5] | 54.5±2.6 [34.3–68.2] | 49.0±2.7 [30.1–65.1] | 61.1±2.0 [50.3–72.4] | 41.2±2.8 [21.9–58.3] |

tific literature. Incorporating inequality constraints (Z+S+I and Z+G+S+I) yields further gains across most datasets, again with the exception of cfRNA. The full constraint set (Z+G+S+I) achieves the best performance in the smallest sample regimes, $n = 5$ and $n = 10$, across all datasets. Together, this suggests that each constraint and type of domain knowledge confers complementary benefit.

**Constraints substantially reduce performance variability.** In practice, a model is trained and evaluated on a single train/test split. This being the case, the model's reliability depends on the spread of outcomes it may produce across different training sets. A model whose test performance varies widely across splits may underperform in deployment depending on the specific realization of the training data, even if its average performance is favorable. Constraining the feasible model space reduces variability across training sets but may introduce bias if the LLM-elicited constraints are inaccurate. We examine this potential trade-off in our experiments. As shown in Table 1, LLM-constrained models consistently exhibit narrower 90% inter-percentile ranges (IPRs) than unconstrained and baseline methods, especially in small-sample settings, with the lower end of the IPR improving by 20–30% AUC at $n = 5$. The reduction in IPR is most pronounced for the full constraint set (Z+G+S+I), which yields some of the tightest IPRs across datasets and training sizes for any method, lowering the risk of poor performance. These results underscore a secondary benefit of LLM-derived structure: constraints not only improve generalization, but also enhance reliability under sampling variation in high-$d$, low-$n$ regimes.

**Comparison to correlation-based constraints.** We study whether the data itself can be used to construct effective constraints. We derive constraints from correlations between features and labels in the training set, comparing their performance to those guided by domain knowledge.

We construct correlation-based analogs of each constraint type, including selecting features with the largest absolute feature–label correlation (for Z); grouping features with the highest pairwise feature–feature correlation (for G); assigning sign constraints according to the sign of the feature–label correlation (for S); and imposing inequality constraints by identifying the most strongly correlated features within the selected set (for I). We also include a random baseline that selects constraints uniformly over the space of valid options. For each constraint type, we match the number of constraints used by the LLM to ensure a fair comparison. Corr. ($n = 5$) uses correlations calculated using only the 5 training samples available to the model, and Corr. ($n =$ all) uses all available data to calculate correlations, approximating a best-case scenario with full knowledge of the data distribution. Table 2 summarizes performance across methods.

LLM-derived constraints consistently outperform correlation-based constraints derived from the training set alone, and they often match or exceed the performance of those based on correlations in the full-dataset. Figure 3 illustrates the agreement between LLM-elicited constraints and correlation structure in the data. In general, selected features tend to have higher abso-

Table 3: **Comparison to human-provided constraints.** Test AUC (%) reported as mean ± standard error [5th–95th percentile] over 2000 random splits for the SSI and COVID datasets, for all convex combinations of constraints (columns). We asked a domain expert to specify constraints based only on feature names and task description, either without external assistance ("Human only") or with access to online resources ("Human + online"). LLM-derived constraints (see Table 1 consistently exceed the performance of expert-defined constraints, especially at the smallest sample size ($n = 5$).

| $n$ | Z+G+S+I | Z+G+S | Z+S+I | Z+G | Z+S | Z |
|---|---|---|---|---|---|---|
| | **SSI** (Human only) | | | | | |
| 5 | 38.5±1.0 [25.1–49.0] | 39.3±1.0 [27.3–50.6] | 39.5±1.2 [28.4–55.7] | 55.6±1.7 [30.6–76.8] | 39.9±1.5 [20.7–55.8] | 53.3±1.7 [32.4–75.9] |
| 10 | 39.8±0.8 [27.9–50.2] | 42.1±0.8 [26.8–53.0] | 41.4±0.9 [28.5–56.5] | 57.7±1.1 [38.6–73.8] | 44.2±1.3 [24.7–61.8] | 60.5±1.3 [39.2–79.4] |
| 20 | 41.9±0.6 [32.2–51.7] | 46.2±0.6 [34.9–54.6] | 44.6±0.9 [28.9–56.9] | 62.8±0.9 [46.4–75.8] | 50.5±0.9 [32.5–63.2] | 62.7±1.0 [45.4–78.4] |
| 50 | 45.1±0.0 [45.1–45.1] | 46.7±0.0 [46.7–46.7] | 52.0±0.0 [52.0–52.3] | 65.8±0.0 [65.8–65.8] | 53.3±0.0 [53.3–53.3] | 73.0±0.0 [73.0–73.0] |
| | **SSI** (Human + online) | | | | | |
| 5 | 68.3±0.5 [53.2–78.7] | 67.5±0.6 [49.0–80.2] | 70.3±0.5 [53.3–80.9] | 56.0±0.7 [31.9–76.4] | 68.3±0.7 [44.7–83.6] | 58.9±0.8 [34.5–79.3] |
| 10 | 70.2±0.4 [56.9–79.7] | 69.9±0.5 [53.0–82.2] | 71.4±0.4 [54.1–82.1] | 59.5±0.6 [38.5–76.6] | 70.7±0.5 [51.3–84.2] | 61.4±0.6 [37.4–80.8] |
| 20 | 72.8±0.2 [62.8–78.6] | 71.7±0.4 [54.6–80.6] | 74.5±0.3 [62.5–82.2] | 62.0±0.5 [41.6–76.3] | 72.8±0.4 [55.9–84.0] | 64.7±0.6 [41.4–81.6] |
| 50 | 75.7±0.0 [75.7–75.7] | 74.0±0.0 [74.0–74.0] | 80.3±0.0 [80.3–80.3] | 71.4±0.0 [71.4–71.4] | 78.6±0.0 [78.6–78.6] | 75.0±0.0 [74.7–75.0] |
| | **COVID** (Human only) | | | | | |
| 5 | 68.3±0.2 [65.3–70.7] | 68.7±0.2 [65.7–71.4] | 67.4±0.3 [63.5–70.6] | 67.1±0.5 [60.7–71.4] | 66.5±0.4 [59.5–70.2] | 64.4±0.7 [51.1–70.7] |
| 10 | 68.7±0.1 [66.2–70.5] | 69.2±0.2 [66.5–71.2] | 67.9±0.2 [65.2–70.3] | 68.6±0.2 [65.1–70.8] | 68.3±0.2 [65.3–71.2] | 67.5±0.3 [59.7–71.0] |
| 20 | 69.4±0.1 [67.6–70.7] | 70.0±0.1 [68.6–71.4] | 69.1±0.1 [66.9–70.7] | 69.7±0.1 [68.5–71.2] | 69.5±0.1 [66.8–71.2] | 69.2±0.1 [67.4–70.7] |
| 50 | 69.6±0.0 [69.0–70.2] | 70.3±0.0 [69.8–70.8] | 69.8±0.0 [69.0–70.4] | 69.9±0.0 [69.4–70.5] | 70.3±0.0 [69.5–71.0] | 70.1±0.0 [69.3–70.6] |
| | **COVID** (Human + online) | | | | | |
| 5 | 66.2±0.2 [63.3–68.3] | 67.3±0.2 [62.8–70.2] | 65.8±0.2 [62.4–68.2] | 66.8±0.4 [59.3–70.5] | 65.9±0.3 [61.2–70.0] | 64.7±0.6 [53.7–70.7] |
| 10 | 67.1±0.1 [64.9–68.8] | 68.6±0.1 [66.7–70.7] | 66.4±0.2 [63.6–68.7] | 67.8±0.5 [59.4–70.8] | 68.0±0.2 [64.6–70.3] | 67.9±0.4 [64.8–71.2] |
| 20 | 67.6±0.1 [66.4–68.6] | 69.4±0.1 [67.8–70.5] | 67.4±0.1 [65.8–68.7] | 69.6±0.1 [67.7–70.7] | 69.1±0.1 [67.3–70.7] | 69.1±0.1 [66.5–70.8] |
| 50 | 67.8±0.0 [67.4–68.3] | 69.9±0.0 [69.3–70.3] | 67.9±0.0 [67.1–68.5] | 70.0±0.0 [69.5–70.6] | 69.9±0.0 [69.2–70.3] | 70.0±0.0 [69.3–70.6] |

lute correlation with the label, grouped features are more mutually correlated, sign constraints match correlation direction, and features marked important exhibit larger signed correlations. However, not all constraints follow these trends. Some LLM-elicited constraints disagree with correlations, which may reflect gaps in the LLM's training corpus, noise in the data, or a strategy that optimizes constraints jointly rather than in isolation. Notably, even when such mismatches occur, an example being a sign constraint that contradicts the correlation direction, performance remains competitive. These findings support the view that though LLM-provided constraints broadly agree with empirical correlation, they can deviate, and that perfect agreement is not necessary for performance gains.

**Comparison to human-provided constraints.** We ask a domain expert, a university faculty member specializing in proteomics, biochemistry, and molecular biology, to provide constraint sets Z, G, S, I using only the feature names and task description, either unaided or with access to any online resources, including scientific publications, UniProt, and publicly available biomedical databases. Table 3 reports test AUC using expert-elicited constraints for the SSI and COVID datasets.

LLM-derived constraints consistently outperform the unaided and online-aided expert across nearly all combinations of constraints and training sizes. Online access improves expert performance on SSI, but not COVID. The human expert provided fewer group and inequality constraints, both with and without access to online resources, compared to the LLM. This may reflect a more conservative elicitation strategy by the expert, or the LLM's capacity to reason about a larger number of relationships. This

suggests an LLM can surpass the ability of a domain expert in designing constrained model, even when the expert is given time and access to online resources. This highlights the potential for LLMs to serve as an alternative or supplement to human expertise in designing sample efficient models.

**Robustness to LLM choice, prompt variation, and regularization strength** Our approach has a single tunable hyperparameter: the L2 regularization strength $\lambda$. All constraint sets are specified by the LLM without user-specified hyperparameters. In the small-sample setting, we fix $\lambda = 10^5$ for all experiments, without cross-validation due to the small size of the training sets. This value was selected in preliminary experiments on the COVID dataset, prior to evaluating other datasets.

## 4 RELATED WORK

**Learning from small-sample, high-dimensional data.** In small-$n$, high-$d$ settings, the standard strategy is to use simple models, often linear, to avoid overfitting (Bühlmann & Van De Geer, 2011). Yet as we show, even strongly regularized models fail to generalize in the extreme data-limited regime we study. Another approach is expert-designed models, where the model class, features, regularization structure, and other inductive biases are handcrafted, yielding performance reflective of expert knowledge, potentially without any data (Domingos, 2012). However, this is resource-intensive and can be infeasible when reasoning over large and complex feature sets is required (Kohavi & John, 1997). Feature selection methods, including methods such as LASSO (Tibshirani, 1996a), Ridge (Hoerl & Kennard, 1970a), RFE (Guyon et al., 2002a), MRMR (Ding & Peng, 2005a), and mutual information (Lewis, 1992), are also widely used to improve generalization. However, these require evaluating alternative models on validation data, inflating sample complexity. This limits applicability to the extreme regime we study, and all of these methods substantially underperform in our benchmarks.

**LLMs for linear model design.** LLM-aided design of linear models is an emerging area of focus, given the central importance of linear models in many real world applications. LLM-Lasso incorporates LLM-informed weights into L1 regularization (Zhang et al., 2025), while LLM-Select uses LLMs for feature selection (Jeong et al., 2024). We benchmark against both and substantially outperform them. LLM-Lasso is highly sensitive to regularization strength, requiring validation and inflating sample complexity, limiting generalization in the sample-limited setting we study. LLM-Select imposes only zeroing constraints, failing to capture richer domain knowledge that can be elicited from an LLM. In contrast, our work targets the underexplored extreme small-sample regime, frames LLM-guided model design as hypothesis class restriction, examines a broad family of composable constraints with complementary benefits, and achieves substantially larger accuracy gains than prior LLM-based methods.

## 5 DISCUSSION

Our results demonstrate that domain knowledge elicited from large language models can be systematically translated into constraints that deliver substantial performance gains in high-dimensional, small-sample settings. Across four clinical datasets, models constrained by LLM-derived domain knowledge achieve improvements exceeding 20% AUC in some cases, with models trained on as few as five examples outperforming baselines that require ten times more data. All other approaches, LLM-based as well as conventional methods, substantially underperform across all datasets, underscoring the absence of effective approaches in the extreme sample-limited regime. In addition, our techniques improve model reliability, where training on few samples can otherwise be highly unstable: the interpercentile range of test AUC improved by more than 30% in some settings. In comparison, constraints from a human domain expert achieved marginal gains, likely reflecting the limited ability of a single expert to evaluate all possible features and constraints in the absence of a large supporting corpus. These results highlight that systematic domain knowledge elicitation from LLMs can fundamentally shift the data requirements of accurate and reliable prediction.

The magnitude of performance gains suggests that LLM-guided constraints open new avenues for improved prediction in critical applications. This includes reducing the costs of patient recruitment and data collection in clinical studies, and enabling predictive modeling for rare diseases and phenotypes where patient populations are inherently limited. Advancing theoretical understanding of why certain constraint types remain robust to LLM misspecification, developing and refining LLM prompts and elicitation strategies, and designing frameworks that integrate LLM guidance with human expertise are key directions for future work.

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

## A  EXTENDED EXPERIMENTAL RESULTS

### A.1  LLM CONSTRAINTS IN LARGER SAMPLE SETTINGS

Figure 4 extends the analysis of Figure 2 by examining regimes where the training set is larger ($n = 20$ and $n = 50$). Full results across all baseline methods for $n = 20$ or $n = 50$ are in Table 1. Each panel reports mean ROC curves over 2000 random train/test splits for logistic models trained under five constraint regimes: Ridge (no constraints), Z only, Z+G, Z+G+S, and Z+G+S+I. Shaded areas denote $\pm$ one standard error of the mean. Columns correspond to the four clinical omics datasets (SSI, COVID, cfRNA, Labor).

Overall, the qualitative trends mirror those observed in the small-sample experiments (Figure 2), though the gains from constraints are attenuated. In particular, constrained models continue to outperform the unconstrained baseline across most datasets, and performance generally improves as additional constraint types are layered. An exception is the cfRNA dataset, where unconstrained Ridge regression surpasses the constrained variants at larger $n$. Further, the advantage of LLM-elicited structure diminishes as more labeled data becomes available, consistent with the intuition that constraints are most valuable when data alone is insufficient to guide the model.

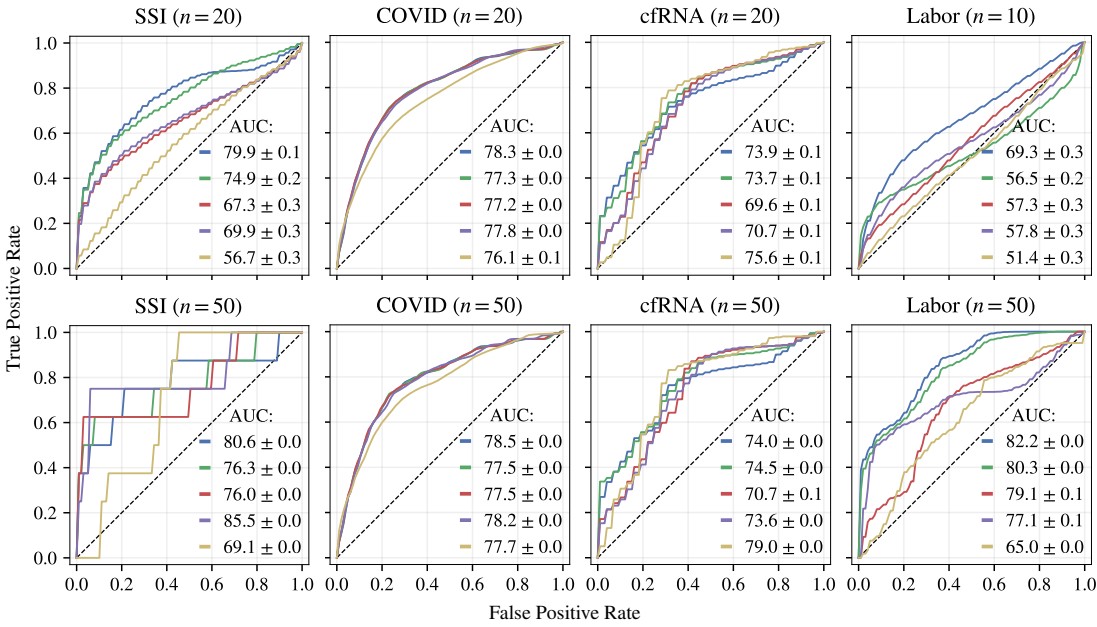

Figure 4: **LLM constraints in larger sample settings.** Each panel shows the mean ROC curve (shaded area: $\pm$ standard error over 2000 random train/test splits) for logistic models trained under five constraint regimes. **Columns:** Four clinical omics datasets (SSI, COVID, cfRNA, Labor). **Top row:** $n = 5$ training examples. **Bottom row:** $n = 10$. Models include: ▨ Ridge (no constraints), ▨ Z only, ▨ Z+G, ▨ Z+G+S, ▨ Z+G+S+I. LLM-derived constraints continue to improve performance across most datasets, with each additional constraint typically yielding further gains. An exception is cfRNA, for which the unconstrained model outperforms constrained variants. Across datasets, the performance gains from constraints are smaller than in small-sample settings, suggesting that LLM-elicited domain knowledge is less useful as more training data becomes available.

### A.2  EFFECT OF REGULARIZATION STRENGTH

In the main text we reported results at a single $L_2$-regularization strength $\lambda$, but here we show that our conclusions are robust across a broad range of values. Figure 5 plots test AUC as a function of $\lambda$ for each dataset and training set size. Several consistent patterns emerge. First, in the most data-limited settings ($n = 5$ and $n = 10$), constrained models substantially outperform the unconstrained Ridge baseline across essentially all values of $\lambda$. The full constraint set ($Z+G+S+I$) typically achieves the highest AUC, with additional constraint types providing complementary improvements.

Second, the advantage of constraints persists at larger training sizes ($n = 20$ and $n = 50$), though the relative gap narrows as unconstrained models begin to stabilize. Again, one exception is cfRNA, where the unconstrained Ridge baseline exceeds constrained models once $n$ is sufficiently large.

Finally, performance is generally stable with respect to the choice of $\lambda$, with constrained models maintaining strong accuracy across several orders of magnitude. This robustness suggests that LLM-elicited constraints mitigate the sensitivity of model performance to the regularization parameter, reducing the burden of hyperparameter tuning in practice.

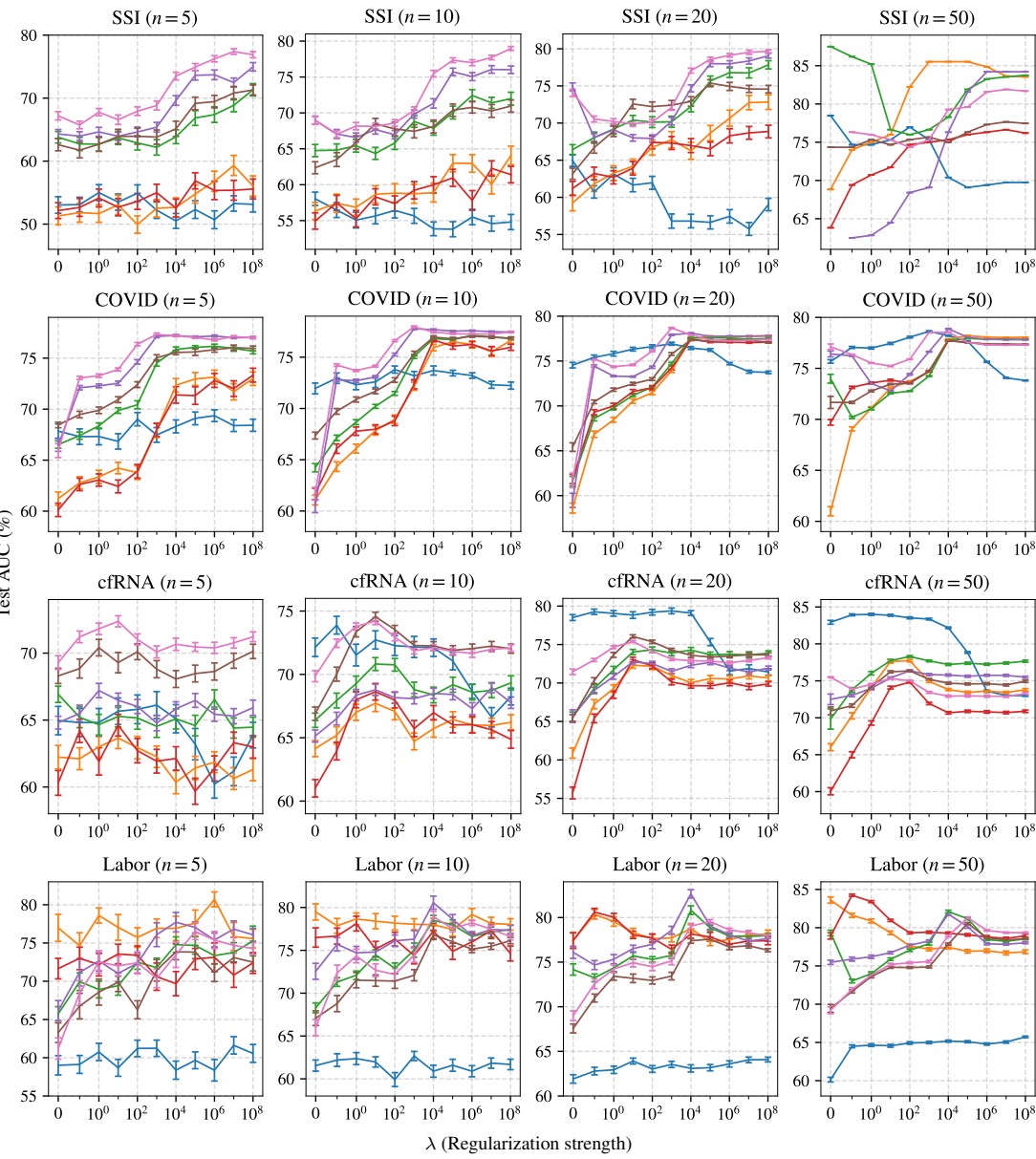

Figure 5: **LLM constraints across L2-regularization strength.** Each panel shows mean test AUC (%) over 200 splits, with error bars indicating standard error, for logistic models trained with different constraint sets across a range of regularization strengths $\lambda$. Models include: ■ Ridge (no constraints), ■ Z, ■ Z+S, ■ Z+G, ■ Z+S+I, ■ Z+G+S, ■ Z+G+S+I. In the most data-limited settings ($n = 5$ and $n = 10$), the full set (Z+G+S+I) generally achieves the highest AUC, and adding constraints tends to improve performance. Performance improvements of the constraints are present across a wide range of regularization strength. On SSI, COVID, and cfRNA, the full constraint set performs best, while on Labor, all constrained models perform comparably.

## A.3 LINEAR BASELINES

Figure 6 reports the performance of a range of feature-selection and LLM-based linear baselines as a function of the regularization strength $\lambda$. Across all datasets and training set sizes, the baselines perform consistently worse than our LLM-constrained models. At high values of $\lambda$, $L_1$-regularized methods (Lasso and Zhang et al.) degenerate by selecting no features, leading to model collapse.

To implement these baselines, we follow the evaluation setup of Zhang et al. For comparability, we generate 200 train/test splits using the same sampling procedure as for our proposed models. For each feature selection method, as well as for the LLM-score approach, we rank up to 160 features. This cap is chosen for computational efficiency and is not restrictive in practice, since the effective number of selected features $d$ never exceeds 100 for any dataset or training size.

Ridge regression baselines are trained with two varying hyperparameters: the number of features $d$ (20 evenly spaced values between 1 and 160) and the regularization strength $\lambda$ (selected from a logarithmic grid). For each value of $\lambda$, we report the performance corresponding to the $d$ that maximizes average test AUC across splits. This procedure yields an optimistic estimate of baseline performance by explicitly choosing hyperparameters based on test outcomes.

LLM-LASSO is handled analogously, with the addition of a second hyperparameter $k$ controlling the strength of the LLM-derived penalty factors. As with the feature-based baselines, we report the best-performing configuration of $k$ for each $\lambda$. Importantly, we also include $k = 0$, which recovers standard LASSO and ensures that LLM-LASSO is lower bounded by its unconstrained counterpart. Thus, in both cases, our evaluation protocol is designed to highlight the maximal achievable performance of each baseline method.

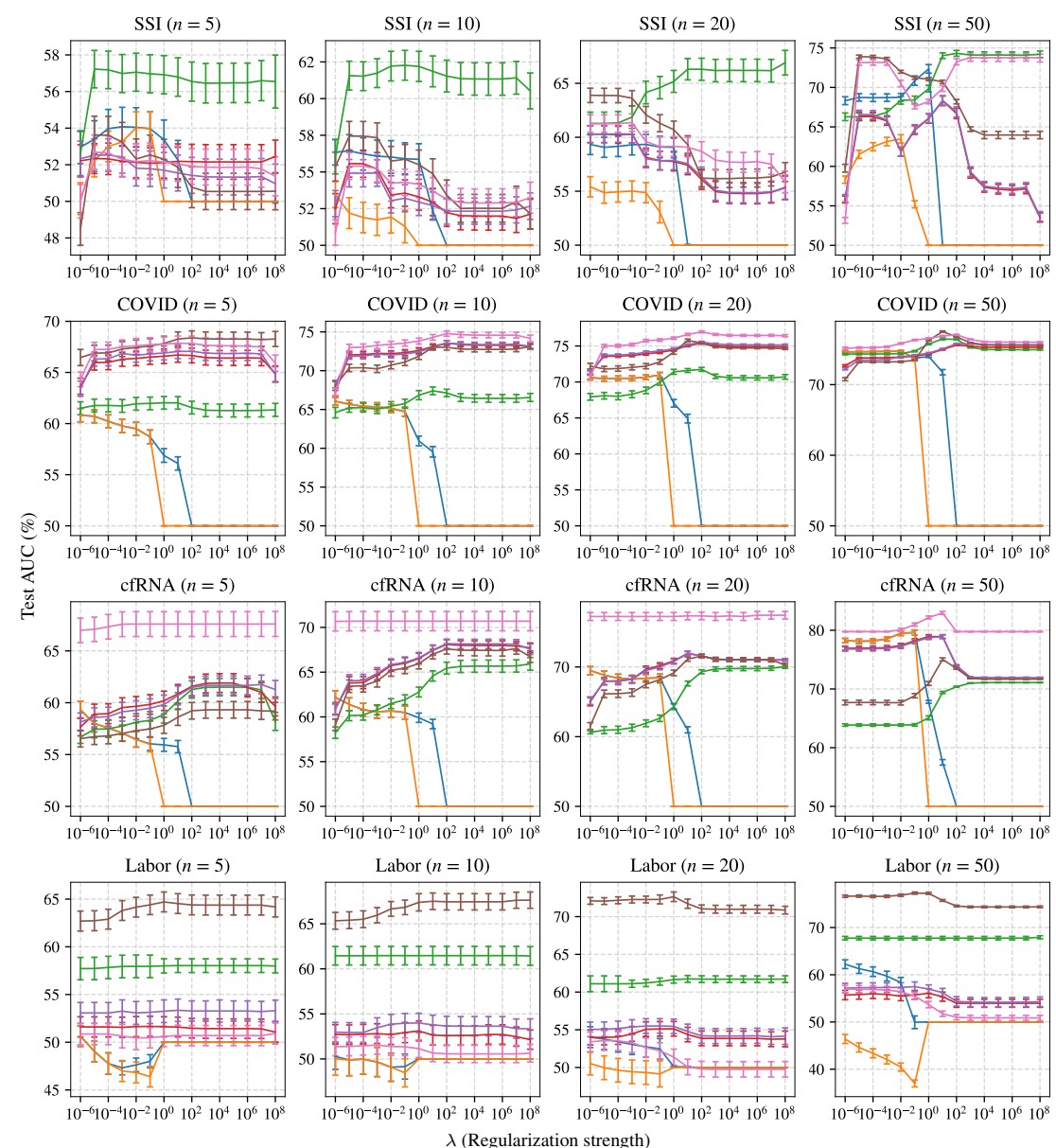

Figure 6: **Baseline methods across regularization strength** Each panel shows mean test AUC (%) over 200 splits, with error bars indicating standard error, for multiple baseline logistic models with different constraints and feature selection methods λ. Models include: ■ Zhang ■ Lasso, ■ Jeong, ■ MI, ■ MRMR, ■ Random Selection, ■ RFE, In the most data-limited settings ($n = 5$ and $n = 10$), our constraint methods consistently outperform all baselines. Note that Zhang and Lasso are the only L1-regularized model and at high regularization strengths, zero features are selected, which leads to model collapse. For feature selection methods, the number of features is treated as a user-defined hyperparameter. We evaluate models across a range of feature set sizes (1-160) and report the best test AUC obtained, providing an upper bound on achievable performance.

## A.4 OTHER MODEL CLASSES

In addition to linear models, we evaluate a range of nonparametric methods and neural networks across the same four clinical prediction tasks. Figures 7 and 8 summarize performance when models are trained either on the full set of molecular features (Figure 7) or restricted to those selected by the LLM (Figure 8). Each bar shows the mean test AUC over 100 random train/test splits, with

error bars indicating the standard error of the mean. Results are reported for training set sizes $n \in \{5, 10, 20, 50\}$.

We consider a diverse set of baselines, including nearest-neighbor methods (KNN-$k$), nearest class centroid (NC), Gaussian, multinomial, and complement Naive Bayes (NB-Gaus, NB-Mult, NB-Comp), decision trees of varying depth (DT-$dX$), unpruned trees (DT-dmax), random forests with $X$ trees (RF-$X$), and multilayer perceptrons with $X$ hidden layers of $Y$ ReLU units (MLP-$XhY$). These models span a wide range of hypothesis classes, from low-capacity generative classifiers to highly expressive feedforward neural networks.

Overall, we find that no nonparametric or neural model consistently outperforms LLM-constrained linear models in the low-$n$ regime. Neural networks and tree-based models show modest improvements at larger training sizes ($n = 20$ or $50$), but remain unreliable when $n$ is small. Multilayer perceptrons consistently underperform linear methods, reflecting the difficulty of training high-capacity models in the data-limited regime. One important note is that on the Labor dataset, performance declines as $n$ increases. This is explained by the training data only containing two positive-class cases, leading to larger subsmples suffering from worse class imbalance.

These results highlight a consistent pattern: in high-dimensional, small-sample tasks, simple linear models, and specifically those augmented with LLM-derived constraints, substantially outperform both classical nonparametric approaches and modern neural networks.

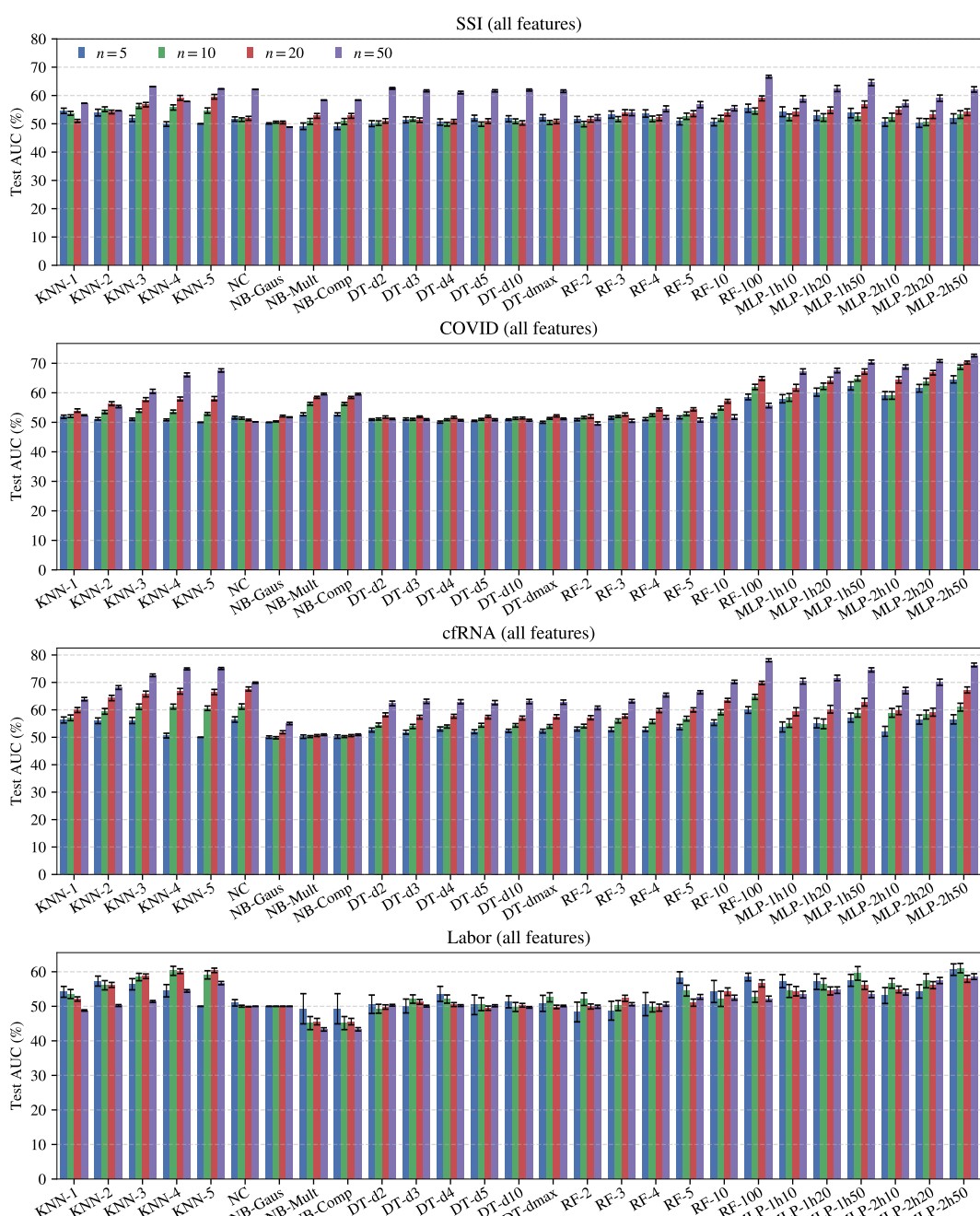

Figure 7: **Nonparametric methods and neural networks trained on all features.** Each panel reports mean test AUC for a range of classifiers across four clinical prediction tasks, using the full set of molecular features. Bars show average AUC over 100 splits for different $n$, with error bars indicating standard error of the mean. Methods include: **KNN-k**, nearest neighbors retrieval with $k$ neighbors; **NC**, nearest class centroid; **NB-Gaus**, Gaussian Naive Bayes; **NB-Mult**, Multinomial Naive Bayes; **NB-Comp**, Complement Naive Bayes; **DT-dX**, decision trees of depth $X$; **DT-dmax**, unpruned trees; **RF-X**, random forests with $X$ trees; and **MLP-XhY**, multilayer perceptrons with $X$ hidden layers of $Y$ units and ReLU activations. The Labor dataset contains only two positive-class examples in the pool from which training sets are sampled, which contributes to the observed decline in AUC as $n$ increases. No single method reliably outperforms linear models in the small-$n$ setting, and all methods consistently underperform LLM-constrained linear models, particularly at low $n$.

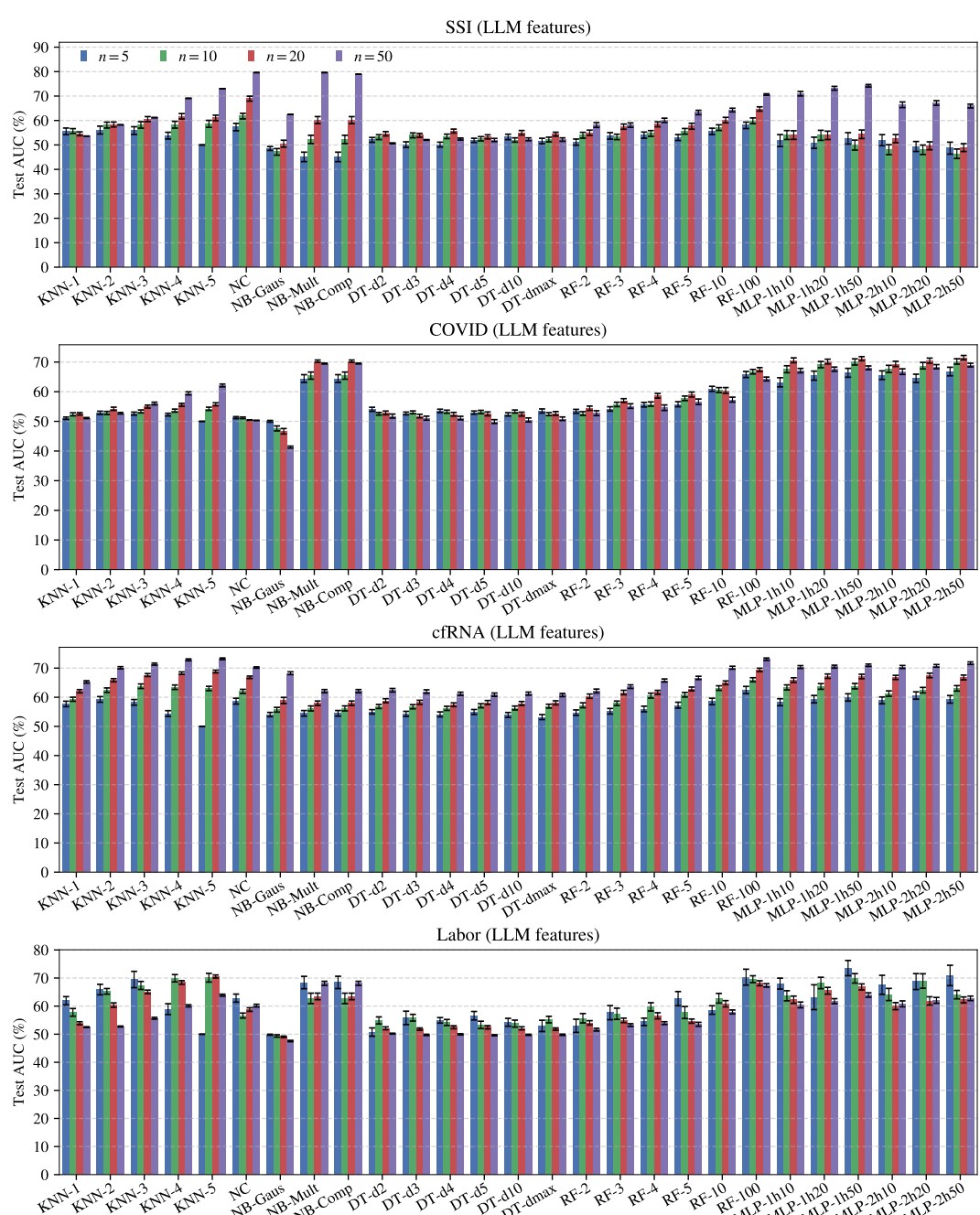

Figure 8: **Nonparametric methods and neural networks trained on LLM-selected features.** Each panel reports mean test AUC for the same set of classifiers as in the previous figure, but limited to features selected by the LLM (i.e., features not in $Z$). Bars show average AUC over 100 splits for different $n$, with error bars indicating standard error of the mean. As before, we note that the Labor dataset contains only two positive-class examples in the pool from which training sets are drawn, which contributes to the observed decline in AUC as $n$ increases. While performance improves compared to using all features, especially for tree-based and nearest-neighbor methods, no method consistently outperforms linear models with additional LLM constraints $G$, $S$, and $I$. This underscores the advantage of incorporating LLM-elicited domain knowledge beyond feature selection.

## A.5 LLM CHOICE

We evaluate models constructed with constraints elicited from multiple large language models (LLMs), including ChatGPT-4 (web interface), GPT-5, Claude 3.5 Sonnet, Gemini 1.5 Pro, and LLaMA-3

70B. Table 4 reports performance of the Z+G+S+I model under $n = 5$ and $n = 10$ training samples across four datasets.

Table 4: **Performance of Z+G+S+I with $n = 5$ and $n = 10$ across datasets.** Entries are Test AUC (%) reported as mean ± standard error with 90% interpercentile range [5th–95th percentile].

| SSI | | | COVID | | |
|---|---|---|---|---|---|
| **Method** | $n = 5$ | $n = 10$ | **Method** | $n = 5$ | $n = 10$ |
| ChatGPT-4 | 76.8±0.3 [62.9–84.9] | 78.3±0.1 [65.9–83.2] | ChatGPT-4 | 78.1±0.0 [75.3–79.0] | 78.3±0.0 [76.1–78.7] |
| GPT-5 | 72.5±0.5 [58–80] | 74.0±0.4 [60–81] | GPT-5 | 74.5±0.4 [69–78] | 72.0±0.5 [66–76] |
| Claude | 70.2±0.6 [55–78] | 75.5±0.4 [61–82] | Claude | 72.8±0.5 [67–77] | 73.5±0.4 [68–77] |
| Gemini | 67.0±0.7 [52–76] | 71.5±0.5 [57–79] | Gemini | 76.0±0.3 [71–79] | 74.2±0.3 [69–78] |
| LLaMA | 69.0±0.7 [54–77] | 70.0±0.6 [56–78] | LLaMA | 70.0±0.7 [64–75] | 71.5±0.6 [65–76] |
| cfRNA | | | Labor | | |
| **Method** | $n = 5$ | $n = 10$ | **Method** | $n = 5$ | $n = 10$ |
| ChatGPT-4 | 71.7±0.1 [59.8–77.1] | 72.9±0.1 [66.5–77.2] | ChatGPT-4 | 67.4±0.4 [61.1–72.2] | 69.3±0.3 [22.1–70.3] |
| GPT-5 | 68.0±0.5 [55–75] | 69.0±0.4 [58–76] | GPT-5 | 61.0±0.7 [55–68] | 65.0±0.5 [59–70] |
| Claude | 62.0±0.8 [50–72] | 67.5±0.5 [56–75] | Claude | 64.0±0.6 [58–70] | 63.5±0.5 [57–69] |
| Gemini | 67.0±0.5 [55–75] | 73.0±0.2 [66–78] | Gemini | 66.5±0.5 [60–72] | 64.0±0.5 [58–70] |
| LLaMA | 65.0±0.7 [52–73] | 66.0±0.6 [54–74] | LLaMA | 59.0±0.8 [53–66] | 61.5±0.6 [55–68] |

## B  OPTIMIZATION AND TRAINING DETAILS

All optimization was implemented in Python 3.10 using the CVXPY package (Diamond & Boyd, 2016). Unless otherwise specified, we used the default ECOS solver with tolerance $10^{-6}$. Results were cross-checked with SCS and OSQP, and no material differences in performance or feasibility were observed. For problems involving both sign and inequality constraints, we explicitly linearized absolute-value terms (e.g., $|w_j| \geq |w_k|$ became $s_j w_j \geq s_k w_k$ when sign information was available) to ensure convexity.

All constraint sets were encoded directly in CVXPY, except zero constraints, where we simply removed features from the train set. Group constraints were implemented by equating weights across feature indices in the same group, $w_j = w_k$. Sign constraints were expressed as linear inequalities of the form $s_j w_j \geq 0$. Inequality constraints were represented as pairwise linear constraints when sign information was available; otherwise, they were left in absolute-value form. Each dataset and training set size was evaluated over 200 random train/test splits. To handle class imbalance, we applied inverse-frequency weights in the loss function, implemented natively in CVXPY.

## C  DATASET AND PREPROCESSING DETAILS

All datasets were from the study of Hédou et al. (Hédou et al., 2024), which documents in detail their preprocessing, which we use there prepcoessed data. For each dataset, we constructed training sets according to the following procedure. For all datasets other than COVID, we first set aside a fixed held-out patient cohort of 50% of the patients, depending on the dataset size. This held-out cohort was reserved exclusively for evaluation and was never seen during training. From the remaining data, we repeatedly generated random training sets of varying sizes $n \in \{5, 10, 20, 50\}$. Each training set was drawn in a stratified manner to ensure that both classes were represented at least once; draws that failed this condition were discarded and resampled.

The COVID dataset was handled differently; an independent cohort from the original study was available and reserved for evaluation. Training sets were generated only from the original discovery cohort, again using stratified subsampling at the specified training sizes.

For each sampled training set, features were standardized by subtracting the mean and scaling to unit variance using statistics computed only from the training data. These scaling parameters were then applied to both the training and corresponding test set to avoid data leakage. No additional normalization, filtering, or imputation was performed.

## D PROMPTS AND ELICITATION DETAILS

We include example prompt templates used to elicit each class of constraint from LLMs, as well as those used in baseline methods. Only dataset-specific fields (e.g., task description and feature names) were modified.

### D.1 ZERO CONSTRAINTS

```
Zero Constraints

You are an expert in biology and medicine assisting in designing an
L2-regularized logistic regression model for a clinical prediction
task.

Dataset:
[Dataset description -- e.g., "The SSI dataset consists of
preoperative blood samples from 91 patients undergoing colorectal
surgery, each with 712 protein abundance measurements. The task is
to predict postoperative surgical site infection (SSI)."]

Features:
[Feature list - e.g. (CRYBB2, VDR, DUSP4, ...)]
[Feature description -- e.g., "Each feature is the
standardized abundance of a specific protein measured from blood."]

Task:
Propose a small subset of the most relevant features to include in
the model. All other coefficients will be fixed to zero. Select
proteins whose biological role suggests an association with infection
risk.

Output format:
- A numbered list of feature names to retain.
- For each, provide a brief justification.
```

## D.2  SIGN CONSTRAINTS

---

**Sign Constraints**

```
You are an expert in biology and medicine assisting in designing an
L2-regularized logistic regression model for a clinical prediction
task.

Dataset:
[Dataset description -- e.g., "The SSI dataset consists of
preoperative blood samples from 91 patients undergoing colorectal
surgery, each with 712 protein abundance measurements. The task is
to predict postoperative surgical site infection (SSI)."]

Features:
[Feature list - e.g. (CRYBB2, VDR, DUSP4, ...)]
[Feature description -- e.g., "Each feature is the
standardized abundance of a specific protein measured from blood."]

Task:
For each selected feature, assign an expected sign (+ or -) to its
regression coefficient, indicating whether higher values of this
feature should increase (+) or decrease (-) with the outcome.

Output format:
- A two-column table: (Feature, Expected Sign).
- For each, provide a brief justification.
```

---

## D.3  GROUP CONSTRAINTS

---

**Group Constraints**

```
You are an expert in biology and medicine assisting in designing an
L2-regularized logistic regression model for a clinical prediction
task.

Dataset:
[Dataset description -- e.g., "The SSI dataset consists of
preoperative blood samples from 91 patients undergoing colorectal
surgery, each with 712 protein abundance measurements. The task is
to predict postoperative surgical site infection (SSI)."]

Features:
[Feature list - e.g. (CRYBB2, VDR, DUSP4, ...)]
[Feature description -- e.g., "Each feature is the
standardized abundance of a specific protein measured from blood."]

Task:
Identify groups of features that should be constrained to have
identical weights in the regression model. Each group should
represent features that are redundant measurements of a shared
factor.

Output format:
- A list of groups (each group is a set of feature names).
- For each, provide a brief justification.
```

---

## D.4 INEQUALITY CONSTRAINTS

```
Inequality Constraints

You are an expert in biology and medicine assisting in designing an
L2-regularized logistic regression model for a clinical prediction
task.

Dataset:
[Dataset description -- e.g., "The SSI dataset consists of
preoperative blood samples from 91 patients undergoing colorectal
surgery, each with 712 protein abundance measurements. The task is
to predict postoperative surgical site infection (SSI)."]

Features:
[Feature list - e.g. (CRYBB2, VDR, DUSP4, ...)]
[Feature description -- e.g., "Each feature is the
standardized abundance of a specific protein measured from blood."]

Task:
Identify the most important features that should be assigned
coefficients with magnitude at least as large as all other features.
These represent the strongest predictors.

Output format:
- A list of important features.
- For each, provide a brief justification.
```

## D.5 LLM-BASED BASELINE METHOD PROMPTS

For these baseline methods, it was impractical to request scores for the full set of hundreds or thousands of features in a single prompt. Instead, we presented the LLM with batches of 50 features at a time, where each batch was drawn sequentially from a list of all features that had been randomly shuffled beforehand. The text feature_list in the prompt template was replaced with the current batch. Aside from this batching procedure and the dataset-specific task descriptions, the prompts are identical to those used in the original papers.

## LLM-LASSO

**Context**: We wish to build a machine learning model that can accurately predict whether a pregnant woman will have Pre-Eclampsia, given cfRNA data collected before clinical diagnosis. Prior to training the model, we first want to identify a subset of the 37184 features (given in batches of 50) that are most clinically important for reliable prediction of the target variable.

**Task**: Provide penalty factors for each of the features. These penalty factors should be integers between 2 and 5 (inclusive), where:
2 indicates a feature strongly associated with the target variable (i.e., it should be penalized the least by Lasso).
5 indicates a feature with minimal relevance to the target variable (i.e., it should be penalized the most by Lasso).

Base these penalty factors on general feature relevance related to the target variable. Do not state that it is impossible to determine exactfactors without further data; instead, provide your best estimate from domain knowledge about each feature's importance in predicting this target.

**Instructions**:
1. You will receive a list of features: {feature_list}.
2. For each feature, produce an integer penalty factor from 2 to 5.
3. List the features and their penalty factors in the exact same order they appear in the list.
4. For each penalty factor, provide a brief statement of how you arrived at that score or why the feature is more or less relevant to the target variable.

**Your Response**:
Must contain one entry per feature, each entry including:
- The feature name.
- The penalty factor.
- A succinct reasoning for that factor.
Do not include disclaimers about lacking full data; rely on your domain knowledge.

Here is an example output, for a different context:
**Track Score**: 5
Reasoning: Track Score is a composite metric that incorporates multiple factors influencing a song's performance. A high Track Score is strongly correlated with higher Spotify streams, making this feature highly relevant.

The list of features is {feature_list}.

---

**LLM-Select**

We wish to build a machine learning model that can accurately
predict whether a pregnant woman will have Pre-Eclampsia, given
cfRNA data collected before clinical diagnosis. Prior to training
the model, we first want to identify a subset of the 37184 features
(given in batches of 50) that are most clinically important for
reliable prediction of the target variable.

Formatting: Each importance score must immediately follow the
corresponding feature name after a double-asterisk-colon (**:**)
and adhere to the following format:
1. For each feature, produce an importance score from 0 to 1.
2. List the features and their importance scores in the exact same
   order they appear in the list.
3. For each importance score, provide a brief statement of how you
   arrived at that score or why the feature is more or less relevant
   to the target variable..

Here is an example output, for a different context:
**Track Score**: 0.93
Reasoning: Track Score is a composite metric that incorporates
multiple factors influencing a song's performance. A high Track
Score is strongly correlated with higher Spotify streams, making
this feature highly relevant.

Provide a score and reasoning for features "{feature_list}" formatted
according to the output schema above:.

---

