# OpenReview forum: "Learning from Few Samples with Language-Model Guidance"
_ICLR.cc/2026/Conference — Submitted to ICLR 2026_

### Official Review · Reviewer_NXLQ · 2025-10-25

**Soundness:** 3
**Presentation:** 3
**Contribution:** 3
**Rating:** 6
**Confidence:** 3

**Summary:**

This paper explores how large language models (LLMs) can inject domain knowledge into machine learning models trained on extremely small datasets (n = 5–50) in high-dimensional settings (e.g., omics data). The authors propose four types of LLM-elicited constraints, feature selection (Z), group constraints (G), sign constraints (S), and inequality constraints (I), which define a convex feasible region for linear model parameters. These constraints are derived via LLM prompting or human experts and integrated into logistic regression training via convex optimization. Across four clinical datasets (SSI, COVID, cfRNA, Labor), models using LLM-derived constraints significantly outperform unconstrained or correlation-based baselines, often achieving comparable or superior performance using only 5–10 samples.

**Strengths:**

1. The paper introduces a systematic and interpretable framework to incorporate LLM guidance as mathematical constraints on model parameters, rather than as data augmentation or feature embedding. Encoding domain priors as convex constraints provides transparent interpretability, robust optimization, and low variance across train/test splits.

2. On small-sample (n = 5–10) omic datasets, the LLM-constrained models improve AUC by up to 20% over standard baselines and even outperform human experts. Each constraint type (Z, G, S, I) is evaluated individually and jointly, showing complementary benefits and robustness.

3. The inclusion of human experts and correlation-based baselines provides strong evidence for the unique added value of LLMs.

**Weaknesses:**

1. The quality of constraints heavily depends on prompt phrasing and model choice (GPT-4o here). While the paper claims robustness, no systematic prompt sensitivity or LLM ablation analysis is provided.

2. While the authors argue that constraints improve sample efficiency, there is no formal generalization bound or bias–variance decomposition to quantify when or why the constraints help (or harm) performance.

3. The assumption that an LLM can meaningfully rank thousands of genes/proteins or infer sign relationships from names alone may be unrealistic in more complex domains. From the experiments, we can also see that cfRNA and SSI, where the method with all four constraints is not the optimal. This highlights the heterogeneity within all these different domains and datasets. More in-depth analysis, such as data distribution, literature analysis, should be given to when and why the constraints are not so help: beyond just speculation.

4. Although justified by small n, the method’s generalization to nonlinear models (e.g., kernel, deep, or Bayesian) is unexplored. The approach may not scale to broader ML tasks.

**Questions:**

1. How sensitive are the results to incorrect or contradictory LLM-specified signs or groupings? Can the method detect and relax inconsistent constraints? No possibility of having conflicting constraints?

2. How does the optimization scale with the number of features (d > 30,000)? Any empirical runtime comparison? The convex optimization with multiple constraints may become expensive, though this is not analyzed.

3. Could combining human and LLM priors iteratively improve constraint quality and interpretability? How good is LLM at giving correct constraints and interpretation of the context?

---

### Official Review · Reviewer_vLhj · 2025-10-31

**Soundness:** 3
**Presentation:** 3
**Contribution:** 2
**Rating:** 6
**Confidence:** 4

**Summary:**

The authors propose a new framework for learning classifiers from extremely small, high-dimensional datasets (a common situation in clinical omics), by eliciting domain knowledge from LLMs (or experts) and encoding it as constraints in linear models. It studies four complementary constraint types: (i) feature selection, (ii) grouping related features to share weights, (iii) coefficient sign constraints, and (iv) inequality constraints that prioritize “important” features. All constraints are composed into a convex, class-balanced, L2-regularized logistic formulation solvable in CVXPY, enabling efficient optimization even with n=5.
Across four clinical tasks (SSI, COVID-19 severity, cfRNA for preeclampsia, and preterm labor), LLM-constrained models substantially improve generalization in the low-n regime.
With only five training patients, the full constraint set (Z+G+S+I) often beats baselines trained on 1X more data, yielding gains exceeding 20% AUC in some settings. Ablation studies indicate each constraint contributes complementary signals; Z+G+S+I is consistently strongest at n=5–10, though benefits attenuate as n grows. The approach appears to be robust to regularization strength and largely stable across different LLMs, suggesting practicality without heavy hyperparameter tuning.

**Strengths:**

This paper introduces a neat and easy solution for modeling complicated tasks when there are very few samples available. For example, for making a diagnosis for a very rare disease with few training samples.
- The authors extract domain knowledge from LLMs as explicit hypothesis-class constraints (Z, G, S, I) for linear models, making priors concrete and testable, without any need for domain knowledge.
- Constraints are cleanly combined resulting in a convex optimization, which can be reliably solved.
- Across all four datasets, with n=5–10, Z+G+S+I wins, sometimes matching/exceeding unconstrained models trained on 10X more samples and showing >20% AUC improvements in some cases.
- The optimization does not seem to be very sensitive to hyperparameters.
- The proposed framework has less performance variability vs the baselines and notably lifts the lower tail of test AUC at n=5, improving reliability of any single train/test split.
- It outperforms correlation-derived constraints (even when computed with all data) and random selections, showing the value beyond naive statistics.
- The results are competitive with (and often beyond) a human expert, LLM-elicited constraints consistently surpass an unaided/online-aided domain expert across sizes and constraint sets.

**Weaknesses:**

- The method is formulated only for binary, linear classifiers; no nonlinear kernels, trees, or deep models are explored beyond brief low-n notes; so it’s unclear how to port these constraints to richer hypothesis classes.
- Benefits shrink at n=20–50, and on cfRNA the unconstrained Ridge overtakes constrained models at larger n, suggesting the approach is most useful only in the extreme low-n regime.
- Only four omics tasks (three with same-study holds; independent external cohort only for COVID). Generalization to other domains, assay types, or multi-site settings isn’t established. This is a major issue since omics datasets have significant confounder (batch) effects (lab, assay, operator, time of processing, etc) and without mitigating such factors, generalization to new batches will be poor.
- In small-n experiments, $\lambda$ is fixed to 1e5 selected on COVID and reused elsewhere without CV; it is pragmatic, but raises concerns about hidden tuning bias and robustness beyond the reported sweep.
- The paper shows LLM constraints can disagree with empirical correlations; while performance often holds up, there’s no formal safeguard if priors are systematically wrong or adversarial.
- Results emphasize AUC; there’s little about calibration, and PPV/NPV at clinically relevant thresholds.

**Questions:**

- The proposed setup focuses on situations where we have very few labelled examples, e.g., for rare diseases. What about unlabelled samples? If it is cfRNA, we could have hundreds of thousands of cfRNA profiles without clinical labels. The proposed framework would be significantly more useful and practical if it can use unlabelled data + few labelled samples.
- The “batch-iness” of omics data is a real challenge and problem for developing any diagnostics model. There are many features (large d) and few samples (small n), so models can easily overfit, even on held-out data from the same set. How would the authors prevent that? I think It would be helpful to see the PCA plots of the data used in training and testing. Moreover, training linear models on the top 10 PCs can be a good baseline.
- Considering what was discussed in the previous point, I find the lack of meaningful performance improvement from n=5 to 50 in three out of four datasets (all except “Labor”) concerning. What is the reason?
- What are the top 10 features (by weight magnitude) for the n=5 case across the four datasets? I believe the authors could unpack the features a bit more. How can a model trained with only 5 points predict COVID severity with an AUC of 0.78? How do these top features compare to baseline models?
- Why are the baseline models outperforming the proposed framework in cfRNA dataset for larger training sizes? I think apart from the need for batch effect correction, another step of feature conditioning/pre-processing needs to be performed, to remove low-variance, low-expression, low-frequency features.
- Why is correlation used in Figure 3? If $x_i$ is continuous and $y$ is binary, why not report AUC or KS-test p-value? Also, why is `ylim =[0, 5]` for correlation?
- Why is the “Random Selection” performance (Table 2)  so high (76.1 vs 78.1 for LLM) for COVID?

---

### Official Review · Reviewer_tBAW · 2025-11-02

**Soundness:** 2
**Presentation:** 2
**Contribution:** 1
**Rating:** 2
**Confidence:** 4

**Summary:**

The authors investigate the effectiveness of several LLM-informed regularization strategies for linear models on small-sample, high-dimensional datasets from clinical settings. In particular, they consider four different strategies: (i) selecting features based on LLMs; (ii) grouping features with LLMs and imposing an equal-weight constraint; (iii) imposing sign constraints on weights based on LLM predictions of expected correlation; and (iv) imposing weight magnitude ordering constraints based on LLM-based feature importance scores. In other words, they combine simple rules for constraining the weights with LLM-generated notions of feature importance and relevance. They evaluate their approach on four clinical datasets, finding that their approach leads to significant performance improvements over other data-driven and LLM-driven baselines (mostly feature selection methods) and in comparison to settings where the same kinds of constraints are designed by human experts.

**Strengths:**

- The proposed approach shows good empirical performance over several data-driven and LLM-driven baselines in extremely small-sample settings (e.g., n=5) on the datasets considered.
- Experimental protocol is generally valid and comprehensive, accounting for various sources of randomness and variability (especially important in small-data settings) and covering several baselines.
- The writing is generally clear and easy to follow.

**Weaknesses:**

- The technical novelty of the proposed approach is limited. It combines existing LLM-based feature selection approaches (e.g., LLM-Select) with relatively simple convex constraints (e.g., sign constraints, group-wise regularization) on the model parameters. In that sense, the proposed work reads as a straightforward and incremental extension of prior works in this space.
- Evaluations are limited to a relatively small number of datasets, with all of them being from the clinical domain. While the small-sample, high-dimensional setting is well-motivated in clinical settings, evaluations on a larger collection of datasets with coverage over other domains would improve the generalizability of findings.
- In comparisons to human-provided constraints, I think the results should be taken with a grain of salt. I would expect these performances to vary across different experts, due to factors including but not limited to differences in levels of experience, possible subjectivity in determining what feature should be deemed "important", etc. While it is an interesting result that LLM-driven constraints can perform on par with when humans are delegated to design the constraints, without additional validation with a greater number of human subjects, I would not go as far as saying "an LLM can surpass the ability of a domain expert in designing constrained model".
- The paper lacks any discussion of possible failure cases for the proposed methods. While I would expect the most capable (often proprietary) LLMs to generally output reasonable constraints, they can always hallucinate and lead to suboptimal performance, especially in domains where the models do not have sufficient parametric knowledge. It would be good to explicitly discuss some of these issues as limitations.
- The authors mention that the proposed method is robust to prompt variation (in last paragraph of Section 3), but there is no explicit demonstration and/or discussion of how this was tested and how variable the downstream performance was in response to different choices of the prompt. As LLMs are generally highly sensitive to the prompting decisions, this also seems to be an important aspect to touch upon.
- All of the tables in the main text are quite dense and hard to visually parse; the presentation of results can be improved for better readability.

**Questions:**

- In Table 1, in the "zero constraint (Z)" setting, why is it that this approach often performs much better than other LLM-based methods like LLM-Select? Modulo the differences in prompting, I think shouldn't applying only Z reduce the proposed approach to LLM-Select and lead to similar numbers across the board?

---

### Meta-Review · Area_Chair_Tr1C · 2026-01-06

**Summary:**

This paper considers strategies for learning from few samples by eliciting relevant knowledge from LLMs, e.g., in the form of coefficient sign and/or magnitude expectations. There was agreement that this work offers a simple and apparently robust approach to learning linear models from few high-dimensional samples in a specialized domain (like healthcare). That said, the empirical results are somewhat underwhelming in the sense that the benefits shrink considerably above ~20 samples. Moreover, the evaluation is limited insofar as only linear models are considered. Reviewers raised some issues with and questions about the experimental setup that went unaddressed.

**Reviewer Concerns:**

N/A; no rebuttal provided.

**Reviewer Scores:**

N/A; no rebuttal provided.

---

### Decision · Program_Chairs · 2026-01-26

Reject